# Optimal Timing for Cardiac Surgery in Infective Endocarditis with Neurological Complications: A Narrative Review

**DOI:** 10.3390/jcm11185275

**Published:** 2022-09-07

**Authors:** Joan Siquier-Padilla, Guillermo Cuervo, Xabier Urra, Eduard Quintana, Marta Hernández-Meneses, Elena Sandoval, Pau Lapeña, Carles Falces, Carlos A. Mestres, Alfredo Paez-Carpio, Asunción Moreno, José María Miro

**Affiliations:** 1Cardiology Department, Health Research Institute of the Balearic Islands (IdISBa), Hospital Universitari Son Espases, 07120 Palma de Mallorca, Spain; 2Infectious Diseases Department, Hospital Clinic—IDIBAPS, University of Barcelona, 08036 Barcelona, Spain; 3Centro de Investigación Biomédica en Red de Enfermedades Infecciosas (CIBERINFEC), Instituto de Salud Carlos III, 28029 Madrid, Spain; 4Functional Unit of Cerebrovascular Diseases, Institute of Neurosciences, Hospital Clinic—IDIBAPS, University of Barcelona, 08036 Barcelona, Spain; 5Cardiovascular Surgery Department, Hospital Clinic—IDIBAPS, University of Barcelona, 08036 Barcelona, Spain; 6Faculty of Medicine and Health Sciences, University of Barcelona, 08036 Barcelona, Spain; 7Cardiology Department, Hospital Clinic—IDIBAPS, University of Barcelona, 08036 Barcelona, Spain; 8Cardiothoracic Surgery Department, The University of the Free State, Bloemfontein 9300, South Africa; 9Radiology Department, Diagnostic Imaging Center (CDI), Hospital Clinic—IDIBAPS, University of Barcelona, 08036 Barcelona, Spain

**Keywords:** endocarditis, surgery, stroke, cerebral infarction, intracerebral hemorrhage, meningitis and mycotic aneurysm

## Abstract

In patients with infective endocarditis and neurological complications, the optimal timing for cardiac surgery is unclear due to the varied risk of clinical deterioration when early surgery is performed. The aim of this review is to summarize the best evidence on the optimal timing for cardiac surgery in the presence of each type of neurological complication. An English literature search was carried out from June 2018 through July 2022. The resulting selection, comprising observational studies, clinical trials, systematic reviews and society guidelines, was organized into four sections according to the four groups of neurological complications: ischemic, hemorrhagic, infectious, and asymptomatic complications. Cardiac surgery could be performed without delay in cases of ischemic vascular neurological complication (provided the absence of severe damage, which can be avoided with the performance of mechanical thrombectomy in cases of major stroke), as well as infectious or asymptomatic complications. In the presence of intracranial hemorrhage, a delay of four weeks is recommended for most cases, although recent studies have suggested that performing cardiac surgery within four weeks could be a suitable option for selected cases. The findings of this review are mostly in line with the recommendations of the current European and American infective endocarditis guidelines.

## 1. Introduction

Infective endocarditis (IE) is a rare but eventually lethal disease consisting in the infection of the endocardial surface of the heart, most frequently affecting cardiac valves. It has a global incidence of 1.5 to 11.6 cases per 100,000 people/year [1]. In the pre-antibiotic era, the mortality of IE was 100%; however, with the discovery and advances in the field of antibiotics and cardiac surgery, mortality has decreased over the years but still holds at approximately 20% [2]. This high mortality is due to different complications caused by the disease: (a) heart failure and paravalvular complications, which can be refractory to therapy; (b) neurological complications (NCs), which by themselves can be the cause of death, and may even prevent patients from undergoing vital cardiac surgeries; and (c) other complications, such as systemic septic emboli to the spleen, kidney, mesentery or skin (causing infarction or a secondary infection), sepsis and septic shock or renal failure.

NCs are the most frequent and severe extracardiac complications. When symptomatic, they have an incidence of 20–55% of all IE patients [3,4,5,6] and a negative impact on outcome, with an increased morbidity and in-hospital mortality [4,5,7]. The diverse NCs and their frequencies are reviewed in Figure 1 [8] and can be classified into vascular and infectious complications, likewise vascular complications can be either ischemic or hemorrhagic.

Ischemic vascular complications are the main NC in IE, accounting for 70% of all NCs [8], and include ischemic stroke and transient ischemic attack (TIA). Ischemic stroke occurs before the diagnosis of IE in 60% of cases [3]. In IE, ischemic strokes are caused by emboli originating in the valve vegetation. Some factors, such as vegetation characteristics (e.g., size, mobility or location in the mitral and the aortic valves) and *S. aureus* infection, have been linked to a greater risk of embolic events [3], whereas this risk decreases after starting antibiotics [9]. The presence of ischemic stroke worsens prognosis in IE patients [4,7,10].

Hemorrhagic NCs are less common, occur mostly in critically ill patients [11], and are the most serious NC in IE [12]. They usually affect the brain parenchyma, although subarachnoid hemorrhage can also occur. There are three possible causes of hemorrhagic events in IE patients: the hemorrhagic transformation of an ischemic stroke, the rupture of an intracranial infectious aneurysm ([IIA], also known as mycotic aneurysm), and cerebral small-vessel vasculitis (inflammation of small vessels due to septic erosion of the arterial wall), with the last being the most frequent [13]. Hemorrhagic complications have been related to *S. aureus* infection and the use of anticoagulant therapy at the time of presentation [4]. Although it has always been considered the NC with the highest mortality, recent findings suggest that mortality from intracranial hemorrhage might not be higher than mortality from ischemic stroke [14]. When considering the cause of intracranial hemorrhage, small-vessel vasculitis might have higher mortality than ruptured IIAs and hemorrhage after ischemic stroke [14].

Although infectious NCs are also less common than ischemic ones, accounting for 1–20% of cases [6], they are not trivial. The most prevalent are meningitis and brain or spinal abscess. Meningitis can either precede or complicate IE. Lucas et al. [15] found that concomitant meningitis and IE raises the risks of ischemic and hemorrhagic events, entailing a possible impact on the outcome of IE. Brain abscess, which is usually multifocal [13], may also worsen the prognosis of IE.

Besides symptomatic events, IE can also cause clinically silent NCs, which are present in 60–80% of all IE patients according to recent studies [6,16,17]. The most frequently observed events are cerebral microbleeds (distributed in cortical areas), acute ischemic lesions (appearing as multiple ischemic lesions of different ages located in watershed territories), and unruptured IIA [16,17]. These findings are not without consequence because they can change diagnosis or the therapeutic plan in many patients [17]. However, it is unclear how they affect outcome. While some studies point out that there are no differences in outcome between symptomatic and asymptomatic NCs, others found better prognosis in asymptomatic NCs, probably related to a higher rate of valve surgery in these patients [6,18,19].

As already mentioned, cardiac surgery is an essential approach and part of the therapeutic plan in IE. Indeed, it is performed in up to 50% of all cases at any time during evolution [2,3,4,20], and it is the strongest predictor of a good outcome among those patients who require it. The current indications for cardiac surgery are stated by the 2015 ESC and AHA guidelines [21,22], namely: severe heart failure, paravalvular complication (abscess or fistula), prevention of embolic event, and uncontrolled infection. These guidelines also define the optimal timing for each indication, according to the severity and need of a fast resolution. However, this optimal timing for patients with NCs is still unclear [20,23]. On the one hand, there is risk of exacerbating the NC during cardiopulmonary bypass, with non-physiological circulation and heparinization, which may lead to a transformation of ischemic into hemorrhagic stroke, to a worsening of cerebral edema and a decrease of blood flow to the ischemic area. In other words, the neurological injury may need time to recover prior to cardiac interventions [8]. On the other hand, the indication for surgery may be life-threatening. In fact, a significant proportion of these patients have at least one indication for cardiac surgery, and their prognosis is poorer when not operated on [5].

In this review we aimed to evaluate the benefits of cardiac surgery for each group of patients with IE and NCs, to determine the best timing to perform the surgical intervention in each group and to assess newer therapeutic approaches that allow cardiac surgery to be brought forward, if required.

## 2. Methods

A literature search was carried out in PubMed from June 2018 through July 2022 using the following key words in combination and alone: ‘endocarditis’, ‘surgery’, ‘stroke’, ‘cerebral infarction’, ‘intracerebral hemorrhage’, ‘meningitis’, and ‘mycotic aneurysm’. This search was performed through the PubMed thesaurus MeSH. The inclusion criteria were: studies published in English or Spanish, study design (case series, case reports, case-control studies, clinical trials, systematic reviews, society guidelines, or consensus papers), to be published during the study period, and the inclusion of patients from all ages and from all countries with IE complicated by NCs (symptomatic ischemic stroke, transient ischemic attack, intracranial hemorrhage, meningitis, brain or spinal abscess, asymptomatic acute ischemic lesions, cerebral microbleed, or intracranial infectious aneurysm) who underwent early or late cardiac surgery (the studies which compared early vs. late surgery had to define the timing for each group). The exclusion criteria were studies published in languages other than English or Spanish, their publication after July 2022, and the absence of a precise definition of early and late cardiac surgery, when those were compared. The resulting selection comprised observational studies, clinical trials, systematic reviews, and society guidelines that were available open access or by subscription, using our institutional subscription. The references cited in the guidelines and review articles that were relevant to the topic of this work were also analyzed.

The results are organized into four sections according to the four groups of NCs: (1) ischemic, (2) hemorrhagic, (3) infectious, and (4) asymptomatic complications. In each group, the analysis covers the benefits of cardiac surgery, its optimal timing, and new approaches to advancing it.

## 3. Definitions

An ischemic stroke is defined as an episode of neurological dysfunction caused by focal cerebral, spinal, or retinal infarction lasting more than 24 h or until death [24]. A transient ischemic attack (TIA) is a transient episode of neurological dysfunction caused by focal cerebral, spinal, or retinal ischemia without acute infarction [24]. A silent central nervous system (CNS) infarction is defined as imaging or neuropathological evidence of a CNS infarction, without a clinical history of acute neurological dysfunction attributable to the lesion [24]. An intracerebral hemorrhage is a focal collection of blood within the brain parenchyma or ventricular system that is not caused by trauma (including parenchymal hemorrhage after CNS infarction) [24]. A subarachnoid hemorrhage is a bleeding into the subarachnoid space (the space between the arachnoid membrane and the pia mater of the brain or spinal cord) [24]. An intracranial infectious aneurysm (IIA) or mycotic aneurysm is an aneurysm occurring in a normal or atherosclerotic artery and resulting from emboli of endocardial origin [25]. Cerebral microbleeds are small, punctuate hypointense lesions seen in T2 MRI sequences, corresponding to areas of hemosiderin-breakdown products from prior microscopic hemorrhages [26]. Meningitis is an inflammatory disease of the leptomeninges, the tissues surrounding the brain and spinal cord and is defined by an abnormal number of white blood cells in the cerebrospinal fluid [27]. Brain abscess is defined as a focal infection within the brain parenchyma, which starts as a localized area of cerebritis, which is subsequently converted into a collection of pus within a well-vascularized capsule [28]. Neurological deterioration is defined as death due to new or worse cerebral bleed and new or worsened clinical neurological status [29].

Although the timing for cardiac surgery is defined in the 2015 ESC IE Guidelines [21] as emergency (surgery performed within 24 h), urgent (within the first week), and elective procedures (after at least 1–2 weeks of antibiotic therapy), there are however no universal definitions for “early” and “late” cardiac surgery. Each study, therefore, defines these time limits differently, so they will be specified in each case.

## 4. Results

### 4.1. Ischemic Vascular Complications

Several studies have evaluated the benefits of valve surgery in IE patients with ischemic neurovascular complications. Ruttmann et al. [10] performed a prospective cohort study including 61 patients with ischemic stroke and 4 with TIAs who underwent cardiac surgery and were followed-up for 20 years. Four patients with TIA survived and had complete neurological recovery. Age-adjusted perioperative mortality risk for ischemic stroke patients was 1.7-fold that of patients without ischemic event but it was not statistically significant (95% CI: 0.73–3.96, *p* = 0.22). Long term mortality risk for stroke patients was 1.23-fold but it was not significant either (95% CI: 0.72–2.11, *p* = 0.45). Lalani et al. [30] carried out a multinational prospective cohort study with 1552 patients with IE who were followed-up for a median of 5.2 years. Of these, 720 patients underwent valve surgery during the initial hospitalization (46%), and 832 (54%) were treated medically. All patients were stratified into groups, depending on the severity and type of complication. In patients with ischemic stroke (18% of all patients), valve surgery was associated with a significant reduction in mortality compared to conservative treatment (absolute risk reduction [ARR] = 13%, *p* < 0.02).

Regarding the timing of cardiac surgery in patients with IE, ischemic strokes are the most frequently analyzed NCs historically. On the one hand, two retrospective studies [31,32] concluded that cardiac surgery should be performed at least two weeks after the ischemic event (see Table 1). In the older of these two studies, Eishi et al. [31] analyzed 112 patients with IE complicated by ischemic stroke before cardiac surgery and found significantly higher in-hospital mortality rate (*p* = 0.009) in patients operated within four weeks than those operated after four weeks, with also a higher neurological deterioration rate (including death related to neurological injury) when comparing patients operated within one week with those operated after four weeks. Therefore, they recommended delaying cardiac surgery at least four weeks from the ischemic stroke. Those patients were later included in a study performed by Angstwurm et al. [32], who analyzed 7 own patients adding them to 180 retrieved from the literature. They found a higher exacerbation rate of NC in patients operated within two weeks than after two weeks (*p* < 0.001).

On the other hand, ten recent studies [10,33,34,35,36,37,38,39,40,41] concluded that, in these patients, cardiac surgery can be performed within two weeks without further risk of neurological deterioration (see Table 1). Three of these recent multi-center studies [34,35,37], totaling 569 patients with ischemic stroke operated for IE, found that cardiac surgery even performed within seven days entailed no significant differences in terms of neurological deterioration or in-hospital mortality when compared with surgery performed after seven days. Furthermore, Okita et al. [37] found significant differences favoring early surgery vs. surgery after 14 days.

In addition, Wilbring et al. [7] prospectively studied 70 patients with IE and NC who underwent cardiac surgery, 53 of which had ischemic stroke, with 1 year follow-up. Mean time to valve surgery in stroke patients was 8 ± 7 days, and postoperative neurological status experienced deterioration in 5.7% of cases, with no deaths due to NC. Sorabella et al. [42] analyzed 308 patients with IE who underwent cardiac surgery within 14 days from diagnosis: 54 had preoperative ischemic NC and 254 did not. They found no significant differences in 30-days mortality (9.3% vs. 7.1%, *p* = 0.57) or in the rate of new postoperative stroke (9.4% vs. 4.7%, *p* = 0.19) between the two groups, with no differences in 10-year survival either (Log rank *p* = 0.74). Ruttmann et al. [40] retrospectively analyzed 440 patients with IE who underwent cardiac surgery from 1995 to 2018 in one center with a median follow-up time of 9 years, with symptomatic stroke being found in 135 patients (30.7%). They classified these 135 patients into two groups: 93 patients with uncomplicated stroke (defined as single or multiple ischemic lesions) and 42 patients with complicated stroke (defined as the additional presence of meningitis, intracranial hemorrhage or abscess). The median time to cardiac surgery was 8 days among patients without stroke and 4 days among patients with stroke. There were no differences in overall perioperative mortality (defined as 30-day mortality or death during the hospital stay) between the stroke group and the non-stroke group (12.6% vs. 13.1%, *p* = 0.39): however, in patients with stroke, there was a trend towards a higher perioperative mortality in patients with complicated stroke compared with uncomplicated stroke (21.4% vs. 6.5%, *p* = 0.063). When comparing patients with stroke who underwent early surgery (defined as surgery within 4 days after stroke) vs. late surgery (after 4 days), there were no differences in perioperative bleeding rate (0.7% vs. 0%, *p* = 0.96) and full postoperative neurological recovery (78.7% vs. 80.9%, *p* = 0.78). Aggravation of the neurological injury after surgery occurred only in one patient who had a preoperative complicated stroke (0.8%). Small et al. [43] recently published a retrospective single-center cohort of 276 patients with IE who underwent valve surgery and 22 of them (8%) had preoperative ischemic stroke; this last group underwent cardiac surgery at a median time of 7.5 days and overall, there was no significant increase in symptomatic hemorrhage after valvular surgery in patients with ischemic stroke compared to those without (*p* = 0.32).

Zhang et al. [44] carried out a retrospective uni-center study including 183 patients with IE complicated by stroke (ischemic or hemorrhagic) and asymptomatic patients who had cerebral MRI done as preoperative routine imaging, all of them undergoing cardiac surgery. They classified them into early surgery (within 14 days) vs. late surgery (after 14 days) and compared postoperative neurological complications (defined as new ischemic stroke or hemorrhagic stroke, expansion of an existing intracranial hemorrhage and new-onset seizures) between both groups. An ischemic stroke was observed in 134 patients (64 from the early surgery group and 70 from the late surgery group) and they found no differences in postoperative neurological complication rates between early and late surgery groups (10.9% vs. 11%, *p* = 0.97).

Tam et al. [29] performed a metanalysis including 27 observational studies (from 1987 to 2016, some of them included in this review), of patients affected from IE with NC who underwent cardiac surgery, and compared in-hospital mortality and postoperative neurological deterioration rate between early and late surgery. They separated patients with ischemic and hemorrhagic NC and classified studies according to their own definition of early and late surgery into early (<7 or <14 days) vs. late (>7 or >14 days) surgery. In the ischemic NC group, they found a higher risk of perioperative death when patients were operated on earlier [<7 days vs. >7 days: RR 1.76 (1.17–2.64) and <14 days vs. >14 days: RR 1.59 (1.08–2.34)], but there were no significant differences in the postoperative neurological deterioration rate between early and late surgery groups [RR 1.82 (0.90–3.66)]. They thus recommended a delay of 7–14 days for cardiac surgery for patients with ischemic NC; however, they could not adjust their results according to the severity of the neurological injury, since this feature has been reported in few studies.

Finally, two studies have evaluated the outcome of cardiac surgery according to the severity of the ischemic event. On the one hand, García-Cabrera et al. [4] defined ‘mild ischemic event’ as a TIA or a stroke affecting <30% of one cerebral lobe, whereas ‘moderate-severe ischemic event’ was defined when there were detected multiple embolisms or a single embolism affecting >30% of one cerebral lobe. They carried out a retrospective multicenter study including 192 patients with IE complicated by an ischemic event, with a follow-up of one year. Of them, 138 (72%) had small ischemic events, and 54 (28%) had moderate-severe ischemic events. In the first group, 54 of the 138 underwent cardiac surgery, with a neurological exacerbation rate of 11% in patients operated within the first week and of 10% and 27% in those operated on in the second and third weeks, respectively. In the second group, cardiac surgery was performed in 15 of the 54, and mortality was 40% (2/5) in the group operated on within two weeks and 20% (2/10) among those operated on after two weeks. Therefore, they recommended delaying cardiac surgery by at least two weeks in cases of moderate–severe stroke but they found no reason to delay surgery in small stroke.

Similar recommendations can be drawn from Murai et al. [38], who developed a retrospective study including 170 patients with IE and vascular NC (whether ischemic or hemorrhagic) and classified them following the NIHSS score at admission into non-severe (NIHSS ≤ 10) and severe (NIHSS > 10) vascular NC. 137 of these 170 patients had non-severe NC, 80 ischemic NC and 57 hemorrhagic. They defined early cardiac surgery as surgery performed within the first 14 days and conventional treatment as surgery after 14 days or medical therapy alone. They found that patients with non-severe ischemic stroke who underwent early cardiac surgery had lower in-hospital mortality rates and a greater survival rate free from IE-related death (*p* = 0.007) than those with conventional treatment (see Table 1B), whereas patients with severe stroke had poor outcomes regardless of the treatment received (36% in-hospital mortality rate in the early surgery group versus 68% in the conventional treatment group, *p* = 0.08). Finally, Samura et al. [39] developed a retrospective multicenter study including 152 patients with IE and ischemic NC. After propensity score matching, they found no significant differences between the 90 matched patients operated within or after 3 days from diagnosis, with a trend favoring early surgery (see Table 1). Furthermore, one third of the patients operated after 3 days developed exacerbation of infection, heart failure, or other organ dysfunction while waiting for cardiac surgery. However, patients included in this study had ischemic lesions smaller than 2 cm, suggesting that patients with mild ischemic stroke might benefit from early cardiac surgery. 

**Table 1 jcm-11-05275-t001:** Results from different studies favoring late (part A) vs. early (part B) cardiac surgery in patients with infective endocarditis and ischemic neurological complication.

(A)						
Year (Reference) of Study	No. of Patients	Design	Timing of Surgery (No. of Patients)	NC-r	In-M	Statistical Analyses
1995 [31]	111	Retrospective Multi-center	<24 h (11) 2–7 d (16) 8–14 d (12) 15–21 d (10) 21–28 d (19) >28 d (43)	≤7 d: 44.4% 8–15 d: 16.7% 15–28 d: 10.3% >28 d: 2.3%	66.3% 31.3% 16.7% 10% 26.3% 7%	NC-r (≤7 d vs. 15–28 d): *p* = 0.02 In-M (<28 d vs. >28 d): *p* = 0.009
2004 [32]	187 *	Retrospective Single-center + patients from the literature *	<3 d (53) 4–14 d (35) 15–28 d (29) >28 d (70)	19% 29% 7% 0%	NA NA NA NA	NC-r (<14 d vs. >14 d): *p* < 0.001
(**B**)						
**Year (Reference) of Study**	**No. of Patients**	**Design**	**Timing of Surgery (No. of Patients)**	**NC-r**	**In-M**	**Statistical Analyses**
2006 [10]	65	Retrospective Single-center	Early ≤ 4 d (NA) Late > 4 d (NA)	3.2% 0%	NA NA	NC-r: *p* = 0.32
2010 [33]	10	Retrospective Single-center	Early ≤ 14 d (8) Late > 14 d (2)	25% 0%	12.5% 0%	NC-r: *p* = 0.43 In-M: *p* = 0.59
2012 [36]	64	Retrospective Multi-center	Early ≤ 14 d (34) Late > 14 d (30)	5.9% 3.3%	17.7% 10%	NC-r: *p* = 1.000 In-M: *p* = 0.483
2013 [34]	198	Retrospective analysis of prospectively collected data Multi-center	Early ≤ 7 d (58) Late > 7 d (140)	NA NA	22.4% 12.1%	In-M: OR = 2.308 (0.942–5.652)
2015 [35]	253	Retrospective Multi-center	Early ≤ 7 d (105) Late > 7 d (148)	42.9% 37.8%	8.5% 9.5%	NC-r: OR = 1.11 (0.63–1.97) In-M: OR = 0.95 (0.35–2.54)
2016 [37]	118	Retrospective Multi-center	1–7 d (36) 8–14 (20) 15–28 (22) >28 (40)	14% ^†^ 0% ^†^ 10% ^†^ 5% ^†^	5% 5% 13.6% 7%	15–28 d vs. 1–7 d: - NC-r: OR 2.23 (0.53–9.43, *p* = 0.274) - In-M: OR 18.7 (1.4–249.12, *p* = 0.027) ◦ 28 d vs. 1–7 d: - NC-r: OR 1.41 (0.36–5.55, *p* = 0.62) - In-M: OR 10.39 (0.77–140.26, *p* = 0.078)
2017 [38]	80 ^‡^	Retrospective Single-center	≤14 d (40) >14 d (40)	NA NA	5% 25%	In-M: OR 0.16 (0.03–0.78, *p* = 0.01)
2019 [39]	90	Retrospective Multi-center	≤3 d (45) >3 d (45)	2% 4%	2% 16%	NC-r: *p* > 0.999 In-M: *p* = 0.058
2021 [41]	54	Retrospective Single-center	≤2 weeks (27) 2–6 weeks (15) >6 weeks (12)	3.7% 0% 8.3%	11.1% ^§^ 6.7% 8.3%	NC-r: *p* = 0.472 In-M: *p* > 0.999

Abbreviations: OR = odds ratio, HR = hazard ratio, NA = not available, NC-r = post-operative neurological complication rate, In-M = In-hospital mortality. Under ‘Statistical analysis’, data in brackets are 95% confidence intervals. No. of patients = patients undergoing cardiac surgery for IE with ischemic NC. * The 111 patients from Eishi et al. [31] were included in this study. ^†^ Okita et al. [37] defined Composite endpoint 1 as the combination of in-hospital mortality and new cerebral event (ischemic and/or hemorrhagic). The difference between this variable and In-M for each time interval is stated in the NC-r column. ^‡^ The 80 patients from Murai et al. [38] included in this table had non-severe stroke (NIHSS ≤ 10). ^§^ Matthews et al. [41] reported 30-day postoperative mortality, rather than in-hospital mortality.

#### Neurological Intervention for Ischemic Stroke Secondary to IE Prior to Cardiac Surgery (Thrombolysis and Mechanical Thrombectomy)

Some studies have contemplated thrombolysis as a potential treatment for cardioembolic stroke secondary to IE. However clinical guidelines have contraindicated this procedure due to the high risk of post-thrombolytic cerebral hemorrhage [21,45].

Several randomized clinical trials have demonstrated that mechanical thrombectomy (in patients without IE) is an efficient alternative option for stroke, reducing morbidity and mortality [46]. This approach has also been recently used in patients with ischemic stroke secondary to IE. Several case series obtained good outcomes, with neurological improvement after the procedure [47,48,49,50,51,52,53,54,55,56,57]. In three of these cases, cardiac surgery was performed thereafter with good postoperative outcomes and no evidence of CNS bleeding [54,55,56].

Bettencourt et al. [58] developed a systematic review including 52 cases from the literature who underwent the treatment of endocarditis-related acute ischemic stroke with mechanical thrombectomy, thrombolysis or their combination (thrombolysis and mechanical thrombectomy) until April 2019. They found a higher risk of intracranial hemorrhage in patients treated with thrombolysis or combined treatment than with mechanical thrombectomy alone (RR 4.14 for thrombolysis vs. mechanical thrombectomy, *p* = 0.001; RR 4.67 for combined therapy vs. mechanical thrombectomy, *p* = 0.01), as well as a trend for functional independence (according to the modified Rankin scale) (*p* = 0.09) and neurological improvement (*p* = 0.07) in favor of thrombectomy, compared to thrombolysis.

Marnat et al. [59] and Feil et al. [60] each performed two multicenter case-control studies comparing the outcomes of mechanical thrombectomy in patients with stroke due to IE with patients with cardioembolic stroke due to atrial fibrillation (AF) (Marnat et al.: 28 patients with IE and 84 patients with AF; Feil et al.: 55 patients with IE and 104 patients with AF). No differences in terms of intracranial hemorrhage after the procedure and procedural complication were found in either of the two studies. However, both found a lower successful reperfusion rate (defined as a modified treatment in cerebral infarction score or mTICI 2b/3) in IE patients compared with AF patients, only reaching statistical significance in Feil et al. (Marnat et al.: 85.7% for IE patients vs. 95.2% for AF patients, *p* = 0.11; Feil et al.: 74.5% for IE patients vs. 87.5% for AF patients, *p* = 0.039). Finally, in both studies a good functional outcome after discharge was significantly less often achieved in IE patients compared to AF patients, whereas Feil et al. found a higher mortality rate in IE patients compared to AF patients (OR = 4.49, 1.80–10.68, *p* = 0.001).

Despite the limited evidence on the value of thrombectomy in patients with IE, these promising results suggest that thrombectomy can significantly impact the clinical course of patients that develop ischemic NC.

### 4.2. Hemorrhagic Vascular Complications

Intracranial hemorrhage is the most severe NC in patients with IE. Although cardiac surgery is necessary in some cases, it is usually hindered by the hemorrhagic event due to the poor status of the patient and/or the fear of aggravating the bleeding. However, Salaun et al. [14] carried out a single-center cohort study, including 68 patients with intracranial hemorrhage secondary to IE with a median follow-up of 15.9 months. Indication for cardiac surgery was present in 60 of the 68 patients, but 38 of 60 (63%) were operated on after a median time of 34 days from diagnosis. They found that in patients with an indication for cardiac surgery, conservative treatment was associated to higher mortality than surgical treatment (Hazard ratio [HR] 5.9, 95% CI 1.54–28.1, *p* = 0.01). The authors concluded that cardiac surgery seems to be beneficial in this scenario and should be performed whenever necessary.

Few studies have analyzed the outcome in relation to the timing of cardiac surgery. The results are depicted in Table 2 and summarized in Table 3. Three studies [4,31,37] recommended postponing cardiac surgery to at least 3–4 weeks from diagnosis. García-Cabrera et al. [4] found a higher in-hospital mortality rate when surgery was performed within three weeks of the hemorrhagic event, although it was not statistically significant. Eishi et al. [31] found no difference in in-hospital mortality between patients operated on within and after four weeks, but most were operated on after four weeks, and this was also ultimately their recommendation. Gillinov et al. [61] retrospectively studied seven patients with IE and hemorrhagic NC who underwent cardiac surgery. Neither neurological deterioration nor death was observed, although mean time to cardiac surgery was 3.89 ± 1.66 weeks. Okita et al. [37] included 54 patients operated on for IE with hemorrhagic NC with different time intervals and also found a trend favoring late surgery (after 22 days or more), although it was not statistically significant.

However, two recent studies [14,62] have raised the possibility that the risk of performing cardiac surgery within four weeks is actually lower than previously thought (see Table 2). Both included a total of 68 patients with IE complicated by intracranial hemorrhage who underwent cardiac surgery with different timings and with none of them suffering postoperative NRL deterioration. In addition, Murai et al. [38] included 59 patients with non-severe hemorrhagic NC (NIHSS ≤ 10): 25 underwent early cardiac surgery and 34 underwent conventional treatment. They found better outcomes in the early surgery group with lower in-hospital mortality rates (0% vs. 22%) and IE-related mortality rates (4% vs. 25%), suggesting that patients with non-severe hemorrhagic NC may benefit from early cardiac surgery (see Table 3 and Table 4). Kume et al. [63] developed a retrospective study including 25 patients with IE and hemorrhagic NC who underwent cardiac surgery: 17/25 within 14 days and 8/25 after 14 days. When comparing both groups, they did not find significant differences in freedom from cerebral hemorrhage after a 12-year follow-up (Log-rank *p* = 0.904). Diab et al. [64] retrospectively studied 363 patients with IE who underwent cardiac surgery: 34 of these patients had preoperative hemorrhagic NC and most of them were operated within 28 days from the neurological event (21 patients within 7 days; 5 patients within 8–14 days, 3 patients within 15–28 days, 3 patients after 28 days and 2 patients with timing unknown). Patients with preoperative hemorrhagic NC compared to those without it did not have higher postoperative neurological deterioration (OR 1.10, 95% CI 0.44–2.73; *p* = 0.84] or mortality (29% vs. 27%, respectively; *p* = 0.84) rates.

In the metanalysis performed by Tam et al. [29], they found a trend suggesting higher risk of perioperative mortality when patients with hemorrhagic NC underwent early cardiac surgery, although it was not statistically significant [<7 vs. >7 days: RR 1.69 (0.51–5.61); <14 vs. >14 days: RR 1.45 (0.71–2.94); <21 vs. >21 days: RR 1.77 (0.77–4.07) and <28 vs. >28 days: RR 0.63 (0.21–1.85)]. In terms of postoperative neurological deterioration rate, they also found a trend favoring late surgery, but it was not statistically significant either [<7 or 14 days vs. >7 or 14 days: RR 2.34 (1.00–5.48)]. They recommended delaying cardiac surgery in this clinical scenario for at least 21 days.

#### 4.2.1. Interventions for Patients with IE Complicated by Intracranial Hemorrhage Prior to Cardiac Surgery

##### Treatment of Ruptured IIAs

Ducruet et al. [65] analyzed a total of 287 patients with IE and ruptured or unruptured IIAs from 27 different case series and found 2 possible managements for ruptured IIAs: open neurosurgery and endovascular treatment. They concluded that, in patients with ruptured IIAs who need cardiac surgery, endovascular therapy is a suitable option, because it permits rapid institution of anticoagulation therapy following valve replacement and also diminishes the waiting period for the cardiac intervention with good postoperative outcomes, a finding corroborated by later studies [62]. Serrano et al. [66] recently published a retrospective analysis of 31 patients with IE complicated by 62 IIAs, 55 of which received endovascular treatment (the 7 IIAs that did not receive endovascular treatment were unruptured, very small, and implanted in an artery inaccessible to catheterization): 30 of the 55 IIAs were ruptured. A total of 22 of the 31 patients underwent cardiac surgery: 18 of the 22 undergoing cardiac surgery had ruptured IIAs. The median times from the start of antibiotic to endovascular treatment and from the endovascular treatment to cardiac surgery were 8 days (IQR: 2.5–13.5) and 2.5 days (IQR: 1–8.2), respectively. In 9 patients, cardiac surgery was performed within 24 h after endovascular therapy of the IIA. Patients were followed for a median of 2.5 months (IQR: 1–5.5) and no cerebral hemorrhage was observed following cardiac surgery in all patients.

##### Nafamostat Mesylate as Anticoagulation during Cardiopulmonary Bypass

Sakamoto et al. [67] evaluated the benefits of nafamostat mesylate, a synthetic protease-inhibitor that has a potent inhibitory action against coagulation factors (XIIa, Xa) and also an anti-inflammatory effect, in cardiopulmonary bypass of 28 patients (22 ischemic strokes and 6 intracranial hemorrhages) who underwent early cardiac surgery for IE with NCs. In cardiopulmonary bypass, anticoagulation consisted of a combination of low-dose heparin and nafamostat mesylate. Patients with ischemic stroke underwent cardiac surgery at a time of 2.1 ± 1.2 days after ischemic stroke, whereas the timing of surgery for patients with intracranial hemorrhage was 3.6 ± 4.2 days. No neurological deterioration was found after surgery in any patient. Ota et al. [68] reported three additional patients with IE complicated by intracranial hemorrhage who underwent cardiac surgery at days 1, 2, and 4 after the NC, using cardiopulmonary bypass and nafamostat mesylate, with no neurological deterioration secondary to surgery either. In this Japanese experience, anticoagulation with nafamostat mesylate could permit the performance of early cardiac surgery in patients who require it, although there is limited experience in patients with IE, and it is still under investigation.

### 4.3. Infectious Complications

The impact of meningitis in the indication of cardiac surgery has been discussed over the years. Although the evidence about this issue is lacking, it is usually considered that meningitis is not a contraindication for cardiac surgery in patients with IE. Lucas et al. [15] developed a prospective cohort study of 24 patients with meningitis and IE. Of these patients, 11 underwent cardiac surgery and had better survival than those who were not operated on, although it was not statistically significant (90% vs. 57%, *p* = 0.17) due to the small number of patients studied.

The evidence about the best timing for cardiac surgery in patients with meningitis is also scarce. In the experience of Lucas et al. [15], the median time to cardiac surgery after diagnosis was nine days (1–45 days). In six of 10 patients, surgery was performed after finishing the course of antibiotics against meningitis, whereas the other four patients were partially treated with antibiotics when undergoing cardiac surgery. There was no difference in outcome between patients who were completely or partially treated for meningitis prior to cardiac surgery. In Wilbring et al. [7], 9 of 70 patients (12.9%) had meningitis. The timing for surgery was 6 ± 8 days. Seven patients improved their neurological status and two deteriorated, but neurological deterioration did not lead to death in any patient. Angstwurm et al. [32] included four patients with IE and meningitis who underwent cardiac surgery at 2, 4, 7 and 8 days after diagnosis, with no postoperative neurological deterioration.

Few studies have included operated patients with IE and brain abscess and, where they have, the patients were not analyzed separately, preventing conclusive results. A total of three patients with brain abscesses were included in the above-mentioned studies: one patient died due to refractory heart failure 24 h after surgery and the other two stabilized, with no neurological deterioration. According to Gianella et al. [69], the management of brain abscesses may include high-dose intravenous antibiotics alone in high-risk patients, patients with abscesses in deep or dominant locations, with multiple abscesses (usual in IE) or with concomitant meningitis. When an abscess is >2.5 cm in diameter, surgical drainage is recommended [69].

All of the above-mentioned studies on the impact of infectious NCs on cardiac surgery had a small number of patients, so caution should be exercised when extrapolating them to give recommendations.

### 4.4. Asymptomatic Complications

Asymptomatic NCs can affect up to 30% of patients with IE [6,70], a figure that can rise to 70–80% if micro-bleeds are included [16,17,71] or even greater in cases of staphylococcal IE [72].

Selton-Suty et al. [18] retrospectively studied 283 patients with IE who underwent initial neuroimaging (cerebral CT scan and/or MRI) and found 100 symptomatic NCs, 35 asymptomatic NCs and 148 normal neuroimages. The rates of cardiac surgery for each group were 43%, 77%, and 54%, respectively. Patients with asymptomatic NCs had a higher rate of valve surgery than those who were symptomatic (*p* = 0.0005), and these interventions were also earlier (time to heart surgery: 14.7 ± 19.0 days in symptomatic patients, 9.5 ± 7.4 days in asymptomatic patients and 16.1 ± 17.1 days in patients without NC). Compared to the symptomatic group, the asymptomatic group had lower in-hospital (8.6% vs. 42%, *p* = 0.0003) and one-year mortality (8.6% vs. 49%, *p* < 0.0001). According to these findings, the authors suggested a protective role of early surgery guided by systematic neuroimaging results in asymptomatic patients.

Hess et al. [16] prospectively studied 109 patients with IE and no neurological symptoms who systematically received an initial cerebral MRI. Neuroimaging showed abnormalities in 78 patients (71.5%). Acute ischemic lesions (40 patients, 37%) and cerebral microbleeds (62 patients, 57%) were the most frequent lesions. A total of 49 patients (45%) with pathological MRI (29 with cerebral microbleed, 20 with acute ischemic lesion and 15 with both) underwent cardiac surgery (timing not specified) with none presenting neurological deterioration or death postoperatively after a mean follow-up of 20.3 ± 19.4 days. Sorabella et al. [42] included in their study 12 patients with asymptomatic cerebral lesions, with no postoperative neurological deterioration reported.

Finally, Chakraborty et al. [73] retrospectively identified 361 patients with IE diagnosed at their center from 2007 to 2014, 127 of which had cerebral MRI: of 48 neurologically asymptomatic patients who underwent MRI, 29 (60%) had MRI abnormalities. In those neurologically asymptomatic patients who had valve surgery, MRI findings did not impact 6-month mortality (OR 0.91; 95% CI: 0.27–3.11; *p* > 0.99) or good neurological functional status at follow-up (OR 1.50; 95% CI: 0.70–3.21; *p* = 0.29).

#### Neurological Intervention for Asymptomatic NCs Secondary to IE before Cardiac Surgery (Treatment of Unruptured IIAs)

Ducruet et al. [65] found two possible managements for unruptured IIAs: conservative management for most small aneurysms (antibiotics for 4–6 weeks with close subsequent follow-up, on the basis that they resolve with medical therapy) and combined antibiotic and neurosurgical or endovascular intervention for large aneurysms, including small aneurysms that enlarge or fail to regress. In patients who need early cardiac surgery, they recommend considering neurological intervention of all unruptured aneurysms depending on the risk of the procedure, although no specific data on outcomes or timing of cardiac surgery was reported in this group of patients. Kume et al. [63] found IIAs in 14 patients included in their study (9 with preoperative symptomatic intracranial hemorrhage and 5 asymptomatic) and found the presence of IIAs prior to cardiac surgery as an independent risk factor for postoperative intracranial hemorrhage (HR = 12.31, 95% CI 2.619–64.82, *p* = 0.002). Four of these 14 patients underwent direct clipping prior to cardiac surgery, with no postoperative neurological deterioration. In the previously mentioned study by Serrano et al. [66], among the 4 patients with unruptured IIAs that received endovascular treatment before cardiac surgery, no intracranial hemorrhage was observed after surgery. Thus, both recommended considering neurological intervention of all IIAs before heart surgery unless hemodynamic instability required emergent cardiac intervention.

## 5. Discussion

Neurological involvement during the course of infective endocarditis, perhaps the most dramatic extracardiac manifestation of this disease, is unfortunately frequent: approximately a quarter of patients will present either focal signs due to embolisms of material from diseased valves (a phenomenon favored by an added state of hypercoagulability [74]), focal signs due to abscesses or mycotic aneurysms, as well as meningeal syndrome due to hematogenous seeding of the central nervous system or encephalopathy secondary to sepsis.

Clinicians face a real challenge when deciding the optimal management of NCs secondary to IE. The decision becomes more difficult when cardiac surgery is required, because evidence on the best timing is not unanimous. Due to the complexity of this issue, the evaluation and guidance from neurology and neurosurgery within the endocarditis team is mandatory when urgent cardiac surgery is needed [21]. Table 4 summarizes the 2015 ESC and AHA recommendations for the management of NCs with the addition of our expert opinion. 

In patients with ischemic NCs, the available evidence suggests that cardiac surgery is beneficial when indicated, with a significant reduction in mortality compared to conservative treatment [10,30]. When analyzing the optimal timing for surgery, some less recent studies recommended postponing cardiac surgery by at least 2–4 weeks [31,32]. However, more recent studies with larger numbers of patients did not find significant differences in terms of mortality or neurological deterioration rate between patients operated on within or after seven days and therefore they found no reason to delay cardiac surgery when urgently needed [7,10,33,34,35,36]. In case of severe ischemic events, García-Cabrera et al. [4] found patients with large ischemic NC on neuroimage to have a higher mortality when early surgery was performed, whereas Murai et al. [38] reached similar conclusions for patients with severe ischemic NC according to their clinical status (NIHSS ≥ 11). In the light of these results, there is information to recommend not delaying cardiac surgery in patients with ischemic NCs without severe neurological damage (i.e., major ischemic stroke and/or NIHSS ≥ 11) when there is an indication to proceed with urgent surgery (for uncontrolled infection, heart failure, residual embolic risk or high-risk structural lesions, such as cardiac paravalvular abscesses). If there is not an indication to proceed with urgent surgery, it could be reasonable to wait in patients with ischemic NCs.

In patients with large artery occlusion ischemic stroke secondary to IE, intravenous thrombolysis is not recommended by the 2015 ESC IE Guidelines, due to the increased risk of intracranial hemorrhage. However, there is growing evidence of the benefits of mechanical thrombectomy in these cases. Further studies are needed to confirm these promising results, which on top of favoring the neurological recovery of the patients, might carry the additional advantage of allowing less risky early heart surgery in patients who require it.

In intracranial hemorrhage, recent evidence shows that cardiac surgery is beneficial when indicated, with higher mortality when patients are not operated on [14]. Earlier evidence on the timing for cardiac surgery found worse outcomes when operated on within four weeks from the NC and suggested postponing cardiac surgery until this time period has expired [4,61]. The 2015 ESC and AHA Guidelines concur with this suggestion. However, three recent studies with 125 patients reported favorable outcomes in patients with intracranial hemorrhage operated on within the first four weeks [14,38,62]. Of note, Murai et al. [38] considered the clinical severity of the hemorrhagic NC and found good outcomes for patients with NIHSS ≤ 10 undergoing early cardiac surgery (see Table 3 and Table 4). Finally, a recent systematic review and meta-analysis [75] found that, in patients with IE, cardiac surgery within 30 days of an episode of intracranial hemorrhage was not associated with increased mortality but had a higher rate of neurological deterioration [71]. In our opinion, a delay of at least 3–4 weeks seems reasonable, although performing cardiac surgery earlier could be valid for mild hemorrhagic NC in selected patients.

Moreover, intracranial hemorrhage caused by small-vessel vasculitis might have a higher mortality than ruptured IIAs and hemorrhage after ischemic stroke [14], so the optimal timing of cardiac surgery may also be different depending on the mechanism of the hemorrhage. It is worth mentioning the review performed by Venn et al. [76], who proposed an algorithm for preoperative management of patients with IE and hemorrhagic NC, depending on the mechanism and severity of the hemorrhage (Figure 1 in their paper). Further studies with appropriate design and sufficiently powered are needed to solve this controversy and eventually reconsider guideline recommendations. In the meantime, the management of these patients should be individualized and discussed in specialized multidisciplinary units (endocarditis teams).

In patients with intracranial hemorrhage, vascular imaging is recommended to rule out ruptured IIAs [8]. If an aneurysm is confirmed, neurological intervention on the aneurysm with either surgical clipping or endovascular treatment is mandatory [61,62]. This procedure avoids the risk of postoperative neurological deterioration and can reduce the delay to cardiac surgery. In this regard, we agree with the recommendation of the current 2015 ESC IE Guidelines. In addition, there are some reports suggesting a lower risk of postoperative neurological deterioration by the use nafamostat mesylate during cardiopulmonary bypass in patients requiring emergent or urgent cardiac surgery, although larger studies with proper designs are needed to confirm its benefits.

In patients with meningitis secondary to IE, cardiac surgery is beneficial in those with indication [15]. The 2015 ESC and AHA Guidelines do not give any recommendation in this case, but the limited evidence on the optimal timing for cardiac surgery is that it should be performed without delay whenever necessary [7,15,32]. Brain abscesses in IE are usually multiple and can be treated medically, but, if they are large (>2.5 cm), surgical drainage should be performed [69]. The evidence on the impact of cardiac surgery is poor, but our opinion is to proceed with the surgery without delay, if indicated.

It may be possible not to delay cardiac surgery in patients with asymptomatic NC, because there is no increase in the postoperative exacerbation rate or mortality [16,18]. We agree with this approach, in line with the 2015 ESC Guidelines recommendations. In patients with unruptured IIA, endovascular treatment (as stated above) may be beneficial in reducing the waiting period prior to cardiac surgery.

### Timing of Cardiac Surgery According to the Urgency of the Indication and the Operative Characteristics in Patients with Neurovascular Complications

It is noteworthy that for those patients with severe neurological damage who also have an indication of urgent surgery due to the presumed dismal prognosis of the IE-related disease, cardiac surgery may still be beneficial compared to conservative treatment. In other words, the probability of survival with an acceptable neurological recovery must be weighed against the risk of a surely fatal outcome if essential surgical interventions are not carried out. On the contrary, in patients without an indication for urgent surgery, it seems reasonable to delay interventions, particularly in patients with hemorrhagic complications of the central nervous system.

Evidence about the influence of the characteristics of the cardiac surgery needed (e.g., one-valve or multivalve surgery, mitro-aortic reconstruction, duration of the cardiopulmonary bypass, etc.) on the decision of the optimal timing for cardiac surgery and its prognostic impact in each complication is scarce. This factor adds complexity to the decision making and should be discussed along with the urgency of the indication of cardiac surgery mentioned above.

In the end, several factors should be taken into account when managing these patients, so the authors agree to emphasize the vital importance of establishing endocarditis teams with all the specialties involved, in order to give the most appropriate treatment on a case-by-case basis.

## 6. Limitations

As it is difficult (perhaps impossible) to organize a randomized clinical trial on the usefulness of cardiac surgery in these patients, all studies are observational. Each study defined early and late surgery differently, and even the 2015 ESC and AHA Guidelines differ in their definitions of the timing of cardiac surgery, which hinders the drawing of conclusions. All studies included in this review have risk of selection and referral-center biases due to patients being managed at highly specialized centers. This review has a risk of publication bias and risk of heterogeneity among studies included (as was reported in Tam et al. [29]). Early cardiac surgery for patients included in the studies of this review was often performed due to clinical deterioration, which could be a source of bias against early surgery for higher-risk patients. Survivor bias should also be considered for the late-surgery group in all cases. Finally, some studies mixed ischemic and hemorrhagic stroke patients in their analyses, with no separate information of each complication alone, which hindered the classification of their patients in this review.

## 7. Conclusions

In patients with IE, the presence of NC could affect decisions on cardiac surgery. The literature summarized in this review support most of the current recommendations from the ESC and AHA Guidelines on patients with ischemic NCs. Further studies are needed to strengthen the level of evidence and to re-evaluate some recommendations, such as the timing for cardiac surgery in case of intracranial hemorrhage with neurological compromise.

## Figures and Tables

**Figure 1 jcm-11-05275-f001:**
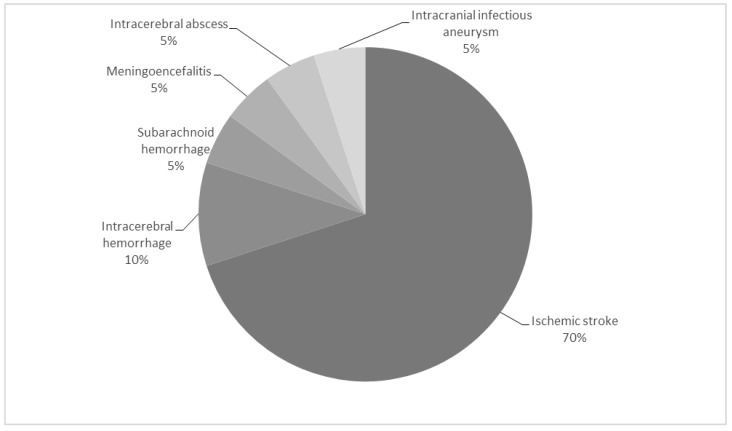
Neurological complication in infective endocarditis and approximate proportions. Adapted from Yanagawa B et al. [8].

**Table 2 jcm-11-05275-t002:** Timing of cardiac surgery and outcomes reported in studies including patients with infective endocarditis and hemorrhagic neurological complication.

Year (Reference) of Study	No. of Patients	Design	Cause of Intracranial Hemorrhage (No. of Patients)	Timing of Surgery (No. of Patients)	NC-r	In-M *
1995 [31]	34	Retrospective Multi-center	NA	<24 h (1) 2–7 d (1) 8–14 d (0) 15–21 d (5) 21–28 d (6) >28 d (21)	100% 0% - 0% 0% 19%	100% 0% - 20% 0% 19%
2013 [4]	12	Retrospective Multi-center	NA	<14 d (4) 15–21 d (3) >21 d (5)	50% 33% 20%	75% 66% 40%
2014 [62]	30	Retrospective Multi-center	Ischemic stroke transformation (4) Ruptured IIA (8) Primary cerebral hemorrhage (10) Subarachnoid hemorrhage (8)	<7 d (5) 8–14 d (6) 15–28 d (9) >28 d (10)	0% 0% 0% 0%	0% 0% 0% 0%
2016 [37]	54	Retrospective Multi-center	NA	1–7 d (13) 8–21 d (17) >21 d (24)	8% ^†^ 11% ^†^ 8% ^†^	15.4% 5.9% 0%
2017 [38]	57 ^‡^	Retrospective Single-center	NA	≤14 d (25) >14 d (32)	NA NA	0% 22%
2018 [14]	38	Retrospective Single-center	Ischemic stroke transformation (16) Ruptured IIA (13) Primary cerebral hemorrhage (9)	<14 d (4) 15–28 d (13) >28 d (21)	0% 0% 0%	NA NA NA
2020 [44]	35	Retrospective Single-center	Intraparenchymal hemorrhage (13) Subarachnoid hemorrhage (26) ^§^	≤14 d (10) >14 d (25)	10.9% 11% ^||^	NA NA

Abbreviations: NA = Not available, IIA = intracranial infectious aneurysm, NC-r = post-operative neurological complication rate, In-M = In-hospital mortality. No. of patients = patients undergoing cardiac surgery for IE with hemorrhagic NC. * There were no significant differences comparing early (<4 weeks) vs. late (>4 weeks) surgery in all studies. ^†^ Okita et al. [37] defined Composite endpoint 1 as the combination of in-hospital mortality and new cerebral event (ischemic and/or hemorrhagic). The difference between this variable and In-M for each time interval is stated in the NC-r column. ^‡^ The 57 patients from Murai et al. [38] included in this table had non-severe stroke (NIHSS ≤ 10). ^§^ 4 patients had both intraparenchymal and subarachnoid hemorrhage. ^||^ Rates from the whole cohort of the study (including ischemic neurological complications). Rates from hemorrhagic complications only were not stated.

**Table 3 jcm-11-05275-t003:** Summary of results reported in the studies listed in Table 2 of patients with infective endocarditis and hemorrhagic neurological complication undergoing cardiac surgery.

Timing of Cardiac Surgery	No. of Patients	NC-r *	In-M ^†^	Statistical Analysi s ^‡^ (<14 d vs. >14 d)
0–14 d	46	3/21 (14%)	4/42 (9.5%)	NC-r: *p* = 0.45 In-M: *p* = 0.22
15–28 d	73	2/41 (5%)	12/60 (20%)	
>28 d	52	4/52 (8%)	4/31 (13%)	
TOTAL	171	9/114 (8%)	13/133 (10%)	

Abbreviations: NC-r = post-operative neurological complication rate, In-M = In-hospital mortality. Okita et al. [37] used different timing intervals, so their patients were excluded from the table; Zhang et al.43 did not state specific NRL complication rates for patients with hemorrhagic neurological complications, so their patients were excluded from the table. * Murai et al. [38] did not state post-operative neurological complication rates, so their patients were excluded from the NC-r column. ^†^ Salaun et al. [14] did not state in-hospital mortality rates, so their patients were excluded from the ‘In-hospital mortality’ column. ^‡^ Statistical analyses were performed using the chi-squared test.

**Table 4 jcm-11-05275-t004:** Management of neurological complications of infective endocarditis according to ESC and AHA current guidelines, compared to author’s own opinion.

	2015 ESC Guidelines [21]	2015 AHA Guidelines [22]	Author’s Opinion
Silent embolism/TIA	No delay in cardiac surgery (class I, LOE B).	No delay in cardiac surgery (class IIb, LOE B).	No delay in cardiac surgery.
Ischemic stroke	No delay in cardiac surgery for heart failure, uncontrolled infection, abscess or persistent high embolic risk, absent coma (class IIa, LOE B). Thrombolysis is not recommended (class III, LOE C).	No delay in cardiac surgery if neurological damage is not severe (class Iib, LOE B). Wait at least 4 weeks in case of major ischemic stroke (class Iia, LOE B).	No delay in cardiac surgery for non-severe ischemic stroke. At least 4 weeks if severe ischemic stroke (NIHSS ≥ 11 [38]) or coma is present. Consider mechanical thrombectomy for major ischemic stroke. Consider proceeding with surgery even if severe ischemic stroke is present in the setting of life threatening hemodynamic or infective disturbances related to endocarditis.
Intracranial hemorrhage	Postpone at least 4 weeks (class Iia, LOE B).	Postpone at least 4 weeks (class Iia, LOE B).	Surgery within 4 weeks may be safe in individualized cases. Consider proceeding with surgery in the setting of life threatening hemodynamic or infective disturbances related to endocarditis.
	IIA should be looked for in patients with neurological symptoms. CT or MR-angiography should be considered for diagnosis. If non-invasive techniques are negative but suspicion remains, conventional angiography should be considered (class Iia, LOE B).	IIA should be looked for in patients who develop severe, localized headaches, neurological deficits, or meningeal signs (class I, LOE B). CT or MR-angiography should be considered for diagnosis. If non-invasive techniques are negative but suspicion remains, conventional angiography should be considered (class IIa, LOE B).	Vascular imaging should be performed to rule out ruptured IIA.
	Neurosurgery or endovascular therapy is recommended for ruptured IIA (class I, LOE C).	-	Endovascular therapy should be performed for ruptured IIA.
Meningitis	-	-	No delay in cardiac surgery.
Brain abscess	-	-	No delay in cardiac surgery. Surgical drainage should be considered for large abscesses.
Cerebral microbleeds and unruptured IIA	-	Cerebrovascular imaging may be considered in all patients, even when no CNS symptoms are present (class IIb, LOE C).	Cerebrovascular imaging may be considered in asymptomatic patients at diagnosis of IE.
	Neurosurgery or endovascular therapy is recommended for very large or enlarging IIA (class I, LOE C). When early cardiac surgery is needed, preoperative endovascular intervention may be considered.	-	Small unruptured IIA can be managed with medical therapy. Absent regression or in enlarging and large IIA, neurosurgery or endovascular treatment should be performed. No delay for cardiac surgery: consider preoperative endovascular treatment of IIA.

Abbreviations: ESC = European Society of Cardiology, AHA = American Heart Association, LOE = level of evidence, TIA = transient ischemic attack, IIA = intracranial infectious aneurysm, NIHSS = National Institute of Health Stroke Score, CNS = central nervous system. Classes of recommendation [21]: Class I: Evidence and/or general agreement that a given treatment or procedure is beneficial, useful, effective. Class II: Conflicting evidence and/or a divergence of opinion about the usefulness/efficacy of the given treatment or procedure, being: Class IIa: Weight of evidence/opinion is in favor of usefulness/efficacy. Class IIb: Usefulness/efficacy is less well established by evidence/opinion. Class III: Evidence or general agreement that the given treatment or procedure is not useful/effective, and in some cases may be harmful. Level of evidence [21]: A: Data derived from multiple randomized clinical trials or meta-analyses. B: Data derived from a single randomized clinical trial or large non-randomized studies. C: Consensus of opinion of the experts and/or small studies, retrospective studies, registries.

## Data Availability

All information supporting the reported results is available from the original sources referenced in the manuscript.

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
