# Peer review of "Optimal Timing for Cardiac Surgery in Infective Endocarditis with Neurological Complications: A Narrative Review"

_jcm, 2022, doi:10.3390/jcm11185275_

Round 1
Reviewer 1 Report
This MS “Optimal timing for cardiac surgery in infective endocarditis with neurological complications: a narrative review.” is to summarize the best evidence on the optimal timing for cardiac surgery in the presence of each type of neurological complication. An English literature search was carried out from June 2018 through February 2022. The resulting selection, comprising observational studies, clinical trials, systematic reviews and society guidelines, was organized into four sections according to the four groups of neurological complications: ischemic, hemorrhagic, infectious and asymptomatic complications. Cardiac surgery could be performed without delay in cases of ischemic vascular neurological complication (provided the absence of severe damage, which can be avoided with the performance of mechanical thrombectomy in cases of major stroke), as well as infectious or asymptomatic complications. In the presence of intracranial hemorrhage, a delay of four weeks is recommended for most cases, although recent studies have suggested that performing cardiac surgery within four weeks could be a suitable option for selected cases. The findings of this review are mostly in line with the recommendations of the current European and American infective endocarditis guidelines.
The study has clinical significance in cardiac surgery.
Major Critiques:
1. There is a major problem with this review. The review nicely identified the indication for cardiac surgery according to the neurological complications. However, what decides whether the cardiac surgery should be performed is not only the neurological complications. There are major concerns about the operative indication: whether the cardiac surgery is urgently needed. Other words, when IE destroyed the heart valves, particularly the left side heart valves (aortic or mitral valve), the mechanical problem of the heart function is so severe that develops acute congestive heart failure so that the urgent valve replacement would sometimes inevitable. In contrast, if the heart valve is slightly dysfunctional the patient can wait for 4 weeks with formal protocol of anti-IE treatment.
2. In view of the above, this reviewer suggests having a separate section to describe the urgency of the cardiac surgery and the type of surgery. It is a balance between the IE and the neurological complications with regard to the indication of cardiac surgery.
Author Response
Major Critiques:
- There is a major problem with this review. The review nicely identified the indication for cardiac surgery according to the neurological complications. However, what decides whether the cardiac surgery should be performed is not only the neurological complications. There are major concerns about the operative indication: whether the cardiac surgery is urgently needed. Other words, when IE destroyed the heart valves, particularly the left side heart valves (aortic or mitral valve), the mechanical problem of the heart function is so severe that develops acute congestive heart failure so that the urgent valve replacement would sometimes inevitable. In contrast, if the heart valve is slightly dysfunctional the patient can wait for 4 weeks with formal protocol of anti-IE treatment.
Answer: We totally agree with the reviewer. Indeed, the recommendations regarding the “tolerable delay” in performing valve surgery in these patients should be interpreted in the context of the urgency that the clinical situation of the patient demands. We have tried to reflect this aspect in various passages of the manuscript (lines 572-578 and lines 634-639 of the “tracked” version and table 4/author’s opinion).
- In view of the above, this reviewer suggests having a separate section to describe the urgency of the cardiac surgery and the type of surgery. It is a balance between the IE and the neurological complications with regard to the indication of cardiac surgery.
Answer: Following the reviewer’s instructions, we have stressed this aspect by adding a new section and a brief comment on this subject (lines 632-641 of the “tracked” version).
Reviewer 2 Report
Dear Authors:
Dear authors:
Very well done. Congratulations. Of course, the subject is extremely complicated. Conflicting evidence everywhere. But you have managed to sort it all out well and give the reader a reasonable overview. I especially like the comparison of the current guidelines with your view derived from the research. My only concern is that the work on hemorrhagic neurologic complications ends in 2018. Is there really nothing to find that is more current? Please check that once again.
Author Response
- Very well done. Congratulations. Of course, the subject is extremely complicated. Conflicting evidence everywhere. But you have managed to sort it all out well and give the reader a reasonable overview. I especially like the comparison of the current guidelines with your view derived from the research.
Answer: Thank you very much for the positive evaluation of our manuscript.
- My only concern is that the work on hemorrhagic neurologic complications ends in 2018. Is there really nothing to find that is more current? Please check that once again.
Answer: Following the reviewer's suggestion, we have extended the bibliographic search for all neurological complications until July 2022, which has allowed us to add 4 new bibliographic references (see references 43, 64, 66 and 75) whose findings we comment on in their corresponding sections (all of them marked up with the “Track changes” function).
Reviewer 3 Report
In general, it is a well written manuscript about an important topic of infective endocarditis. Indeed, patients with infective endocarditis suffer from serious complications such as neurologic complications and the timing of surgical intervention is often critical to perform under these circumstances. My comments for the authors are the following:
1. Please, correct some syntax errors in the manuscript (i.e. introduction).
2. In the section of methods, you should present the inclusion/exclusion criteria in details. Also, discuss the availability of the studies that you included in your presentation.
3. Another suggestion is that the segment of definitions should be incorporated in the introduction along with the interpretation of the neurologic complications of infective endocarditis.
4. Please, provide better presentation of the tables (i.e. include all the abbreviations at the bottom line)
5. In the segment of discussion, you could provide in short some pathophysiologic evidence about infective endocarditis and neurologic complications.
Author Response
- In general, it is a well written manuscript about an important topic of infective endocarditis. Indeed, patients with infective endocarditis suffer from serious complications such as neurologic complications and the timing of surgical intervention is often critical to perform under these circumstances. My comments for the authors are the following:
Answer: Thank you very much for the positive evaluation of our manuscript.
- Please, correct some syntax errors in the manuscript (i.e., introduction).
Answer: We have corrected the syntax errors detected throughout the manuscript.
- In the section of methods, you should present the inclusion/exclusion criteria in details. Also, discuss the availability of the studies that you included in your presentation.
Answer: We have included in the methods section the inclusion/exclusion criteria of our study (see lines 151-162 of the “tracked” version). We have also added the availability of the studies included in our study (see lines 163-164 of the “tracked” version)
- Another suggestion is that the segment of definitions should be incorporated in the introduction along with the interpretation of the neurologic complications of infective endocarditis.
Answer: Thank you for your comment. We believe that academically it is better that the definitions remain in the methods section (see lines 170-197 of the “tracked” version) and for this reason we have not modified this point.
- Please, provide better presentation of the tables (i.e., include all the abbreviations at the bottom line)
Answer: Following your suggestion, we have added the meaning of all abbreviations at the bottom of all tables.
- In the segment of discussion, you could provide in short, some pathophysiologic evidence about infective endocarditis and neurologic complications.
Answer: As suggested, we have added a comment on the pathophysiology of neurologic complications in patients with infective endocarditis in the discussion section (see lines 547-553 of the “tracked” version), adding a new reference (https://doi.org/10.1371/journal.pone.0261429).
Reviewer 4 Report
The authors tried to systematize into 4 groups the most frequent complications arising from infectious endocarditis.
Of course, this is a rather conditional systematization, since there can be a huge difference between the condition of two patients, which we call ischemic neurological complications in both. That's why we can't use the same surgical treatment strategy for both patients.
In the described studies, there is also a large heterogeneity in terms of timing: interventions with a median of 4 days and an intervention delay of up to 2 weeks.
Due to the different indications for cardiac surgery in infectious endocarditis, there may not be an ideal time for cardiac surgery, so we still use a strategy of individual solutions.
The only thing I didn't see in the article was information about the significance of the type, complexity and duration of the proposed operation for choosing the intervention time. That is, if the patient is supposed to have a correction of one valve or a combined intervention, long-term. Will there be a difference in the intervention time?
I think this information will complement this review.
This is a pretty interesting review. In general, I have no serious comments.
This article will be useful not only for cardiac surgeons, but also for a wider range of specialists related to cardiac surgery.
Author Response
- The authors tried to systematize into 4 groups the most frequent complications arising from infectious endocarditis. Of course, this is a rather conditional systematization, since there can be a huge difference between the condition of two patients, which we call ischemic neurological complications in both. That's why we can't use the same surgical treatment strategy for both patients. In the described studies, there is also a large heterogeneity in terms of timing: interventions with a median of 4 days and an intervention delay of up to 2 weeks. Due to the different indications for cardiac surgery in infectious endocarditis, there may not be an ideal time for cardiac surgery, so we still use a strategy of individual solutions. The only thing I didn't see in the article was information about the significance of the type, complexity and duration of the proposed operation for choosing the intervention time. That is, if the patient is supposed to have a correction of one valve or a combined intervention, long-term. Will there be a difference in the intervention time? I think this information will complement this review.
Answer: We truly agree with the reviewer. Certainly, the characteristics of the cardiac surgery needed (e.g., one-valve or multivalve surgery, mitro-aortic reconstruction, duration of the cardiopulmonary bypass, etc.) adds complexity to the decision making and should be discussed along with the urgency of the surgical indication. Unfortunately, this aspect is poorly addressed in the literature. Therefore, we strongly emphasize the vital importance of establishing Endocarditis teams in order to offer the most appropriate therapeutic approach for each patient in a case-by-case decision (see lines 642-651 of the “tracked” version).
- This is a pretty interesting review. In general, I have no serious comments. This article will be useful not only for cardiac surgeons, but also for a wider range of specialists related to cardiac surgery.
Answer: Thank you very much for the positive evaluation of our manuscript.
Round 2
Reviewer 1 Report
The revision considered my previous comments.